# Information and Communication Technologies in Lung Transplantation: Perception of Patients and Medical Teams

**DOI:** 10.3390/pharmacy10040075

**Published:** 2022-06-30

**Authors:** Sébastien Chanoine, Christelle Roch, Léa Liaigre, Matthieu Roustit, Céline Genty, Elisa Vitale, Jean-Luc Bosson, Christophe Pison, Benoît Allenet, Pierrick Bedouch

**Affiliations:** 1CHU Grenoble Alpes, Pôle Pharmacie, F-38000 Grenoble, France; sebastien.chanoine@univ-grenoble-alpes.fr (S.C.); christelle.roch@hotmail.fr (C.R.); lliaigre@chu-grenoble.fr (L.L.); evitale@chu-grenoble.fr (E.V.); ballenet@chu-grenoble.fr (B.A.); 2CNRS, TIMC UMR5525, MESP, Université Grenoble Alpes, F-38041 Grenoble, France; celine.vermorel@univ-grenoble-alpes.fr (C.G.); jlbosson@chu-grenoble.fr (J.-L.B.); 3Inserm CIC 1406, CHU Grenoble Alpes, Université Grenoble Alpes, F-38000 Grenoble, France; mroustit@chu-grenoble.fr; 4CHU Grenoble Alpes, Pôle Thorax et Vaisseaux, Clinique Pneumologie, F-38000 Grenoble, France; cpison@chu-grenoble.fr

**Keywords:** lung transplantation, health information technologies, adherence

## Abstract

Optimal therapeutic management is a major determinant of patient prognosis and healthcare costs. Information and communication technologies (ICTs) represent an opportunity to enhance therapeutic management in complex chronic diseases, such as lung transplantation (LT). The objective of this study was to assess the preferences of LT patients and healthcare professionals regarding ICTs in LT therapeutic management. A cross-sectional opinion survey was conducted among lung transplant patients and healthcare professionals from the French lung transplantation centers. Five ICTs were defined (SMS, email, phone, internet, and smartphone application) in addition to face-to-face communication. An unsupervised approach by Principal Component Analysis (PCA) identified lung transplant patient profiles according to their preferences for ICTs. Fifty-three lung transplant patients and 15 healthcare professionals of the French LT centers were included. Both expected ICTs for treatment management and communication. Phone call, face-to-face, and emails were the most preferred communication tools for treatment changes and initiation. PCA identified four ICTs-related profiles (“no ICT”, “email”, “SMS”, and “oral communication”). “Email” and “oral communication” profiles are mainly concerned with treatment changes and transmission of new prescriptions. The “SMS” profile expected reminders for healthcare appointments and optimizing therapeutic management. This study provides practical guidance to enhance LT therapeutic management by ICT intervention. The type of ICT used should take into account patient profiles to improve adherence and thereby the prognosis. A combination of strategies including information, education by a multidisciplinary team, and reminders is a promising approach to ensure an optimal management of our patients.

## 1. Introduction

With more than 32,000 procedures performed worldwide in the last 30 years, lung transplantation (LT) is a relevant treatment for lung diseases with end-stage chronic respiratory failure, leading to an improvement of patients’ survival and quality of life [1]. LT success was made possible not only by operative technique progress, but also by a multidisciplinary intensive monitoring of patients [2,3], which usually includes lung specialists, thoracic surgeons, nurse coordinators, dieticians, physiotherapists, social workers, and pharmacists [4]. The challenges of the multidisciplinary transplant team are to prevent efficiently perioperative mortality but also the incidence of acute rejection, chronic allograft dysfunction, and infection, which are the major causes of death. Immunosuppressive treatments are the cornerstone of LT therapeutic management. They are characterized by a narrow therapeutic index and many potential drug–drug interactions associated with clinical impact, requiring close pharmacological and clinical monitoring. Because of a wide range of long-term complications (cardiovascular, metabolic, gastrointestinal, neurological, hematological, and infectious diseases), lung transplant patients receive preventive or curative therapies in addition to immunosuppressive therapy, which lead them to take eight medications daily on average [4].

Improving patient education and therapeutic management adherence is essential in LT [4,5]. A significant proportion of transplant patients are non-adherent (20–30%) [6,7,8], and a gradual decline is observed over time [9,10,11,12]. The failure in adherence to immunosuppressive drugs is responsible for graft rejection with consequences on morbidity, mortality, and also direct and indirect healthcare costs [13,14,15,16,17,18,19]. Recent studies on LT consider not only adherence to immunosuppressive drugs, but also all key factors in LT success (i.e., adherence to other drugs, complication clinical signs, breath measurement, dietary measures, physical activity, crowding out of risk factors, and regular medical or blood markers examinations) [6,7,15,16].

The use of information and communication technologies (ICTs) may be an answer to improve lung transplant patients’ management. Developed during the last decades, ICTs combine techniques used in the treatment and transmission of information, mainly telecommunication, electronic, or internet tools [20]. They are increasingly used in health, especially to facilitate information flow between health professionals and patients [21]. A heterogeneous evidence level was shown for the use of SMS (Short Message Service) for preventive health measures (i.e., smoking, overweight and obesity, attendance of medical appointments, etc.) [20,22,23,24,25,26,27]. More broadly, there is not enough evidence to conclude that ICT interventions improve patients’ adherence [28,29,30,31,32]. However, ICT effectiveness has been shown in several chronic diseases, such as asthma, diabetes, or human immunodeficiency virus infection [32,33,34]. Few studies were conducted in transplantation to assess the relevance of ICT use in patients’ management [30]. To our knowledge, only one study was conducted in LT, assessing the acceptance and use of a smartphone application for daily health self-monitoring in a randomized controlled trial [35]. Given the evidence heterogeneity in ICT usefulness and ICT multiplicity, according to patients and health care professionals, potential benefits of these devices have to be defined for the monitoring of lung transplant patients. In this context, the present study aimed at assessing the preferences of lung transplant patients and healthcare professionals regarding ICTs in LT therapeutic management in order to provide practical guidance to enhance this management by ICT intervention.

## 2. Materials and Methods

### 2.1. Study Design

A cross-sectional opinion survey was conducted among lung transplant patients followed at Grenoble Alpes University Hospital (inclusion criteria: aged 18 years or older, speaking and understanding French) and healthcare professionals among the French lung transplantation centers (Nantes, Strasbourg, Paris, Marseille, Toulouse, Lyon, and Grenoble), which are all members of the GETTAM (Cystic Fibrosis Transplanted Adult Education Therapeutics Group).

### 2.2. Information and Communication Technologies

Five ICTs were studied and defined for patients and healthcare professionals as follows [20,21,27]: (1) SMS: the patient receives a text message from an LT center and confirms reading it by sending an acknowledgment; (2) Email: the patient receives an email from am LT center and confirms reading it by sending an acknowledgment; (3) Phone call: the patient receives a phone call from an LT center; in case of no response, a message is left on their answering machine, and the patient has to confirm playback; (4) Internet: the patient can access, by email address and password, a website with general and personalized information; (5) Smartphone application: the patient downloads and installs free software on their smartphone providing access to information modules. Face-to-face communication, which is not a type of ICT and which is defined as oral information provided by a healthcare professional from an LT center during the patient visit, was also studied. ICTs were not already provided for patients and care providers.

### 2.3. Evaluation of Preferences between ICTs

A questionnaire based on a literature review and clinical practices (protocols, communication, information flow) was designed to evaluate preferences of both patients and healthcare professionals regarding ICTs in LT therapeutic management. It was structured into 15 multiple-response questions related to probable use scenarios based on risk situations encountered in LT management (for example, missing drug intake, healthcare appointment, treatment modification) [6,7,15,16] (Appendix A). Response proposals consisted of the five ICTs studied, face-to-face communication, or none of them. The questionnaire was validated by a multidisciplinary team (4 pharmacists, 4 physicians, and 4 nurses). Its readability was confirmed by a Flesch–Kincaid test (i.e., a score at 61.5, meaning standard-level text). The questionnaire was filled by a 15 min interview, conducted by a pharmacy resident, face-to-face (patients) or by phone (patients and healthcare professionals). To facilitate the interpretation, three specific functionalities of ICTs in the risk situations emerged: reminder, transmission, and communication. More specifically, reminder included reminders for daily drug intake, punctual drug intake, optimized therapeutic management (e.g., shifting immunosuppressant intake, vaccinations, measures of breath), and healthcare appointments; transmission included the transmission of new prescriptions and of data such as treatment modification or treatment initiation; and communication was between the patient and the medical team.

In addition, demographic and clinical characteristics (age, sex, marital status, occupational, indication of LT, time after transplantation), LT management data (prescribed immunosuppressive drugs, healthcare appointments, treatment modifications, transmission of new prescriptions, optimization of therapeutic management, and communication), and ICT (SMS, internet, emails) accessibility and use were collected for LT patients.

### 2.4. Statistical Analysis

Categorical data were expressed as a frequency and percentage, and quantitative data were expressed as the mean and range. An unsupervised approach by a Principal Component Analysis (PCA) was used to identify LT patients’ profiles regarding their preferences. As previously performed, the most representative question in a theme was introduced into the model to achieve the PCA because of the high number of questions asked [36]. Statistics were performed using STATA 12.0 (StataCorp, College Station, TX, USA).

## 3. Results

### 3.1. Patients

Among the 69 lung transplant patients followed at the Grenoble Alpes University Hospital, 53 were included in this analysis (Figure 1). The response rate was 100%. The mean age was 51-years-old (range: 19–72), and 40% were women (Table 1). The average time after transplantation was 3.5 years (range: 2 months to 15 years). Immunosuppressive therapies included tacrolimus (98%), corticosteroids (98%), mycophenolate mofetil (85%), everolimus (45%), and azathioprine (7%). Regarding ICT accessibility, 94% of patients had a mobile phone, 70% had a subscription with free and unlimited SMS, and 87% had internet access at home. Patients who were comfortable with SMS or internet and emails were significantly younger than those who self-reported difficulties with these ICTs (median years (Q1; Q3): 52 (37; 63)- vs. 63 (62; 66)-years-old, *p* = 0.01, and 52 (36; 62)- vs. 63 (55; 66)-years-old, *p* = 0.01, respectively).

### 3.2. Patients’ Preferences about ICTs

Most of the surveyed patients reported that they would not like to communicate through ICTs. Some patients expected ICTs for treatment management and communication with healthcare professionals (Figure 2). Phone call and face-to-face were more preferred than the other tools for communication between the patient and the medical team and for treatment modification and initiation. Concerning reminders, around 35% of patients would appreciate reminders for taking punctual medications, mainly by SMS or emails, but 96% would not like any reminder for daily drug intake, including immunosuppressive drugs. Almost 20% would appreciate an SMS to remind them about healthcare appointments. The proposed answer of “smartphone application” was chosen only twice for reminders. Similarly, the proposed answer of “website” was never selected by the patients. Both proposals were not considered for the PCA.

Four ICT-related profiles were identified by PCA (39% of the variance): (1) “no ICT” profile; (2) “email” profile; (3) “SMS” profile; and (4) “oral communication” profile, including phone call and face-to-face communication. The “email” and “oral communication” profiles were mostly found for treatment modifications and transmission of new prescriptions and the “SMS” profile for reminders of healthcare appointments and to optimize therapeutic management. The analysis of the individual’s cloud according to age and to LT indication highlighted that the “no ICT” profile was mainly composed of LT patients for chronic obstructive pulmonary disease, emphysema, or bronchiectasis and aged over 60-years-old, whereas the “email” profile mainly included LT patients with cystic fibrosis and aged under 40-years-old.

### 3.3. Healthcare Professionals’ Preferences about ICT in the Management and Monitoring of LT Patients

Among the 11 French lung transplantation centers, 15 healthcare professionals (9 nurses, 3 physicians, 2 pharmacists, and 1 psychologist) answered the questionnaire. Similar trends were observed between patients’ and healthcare professionals’ preferences about ICTs (Figure 3). Most of healthcare professionals felt that ICT use was not suitable as a reminder for home breath measures (*n* = 9) or daily drug intake, including immunosuppressive therapies (*n* = 12). They considered mostly phone call, SMS, and email as ICTs, according to the situation. Phone call was the preferred method to notify about a treatment modification (*n* = 15) or to communicate with patients (*n* = 12). Emails were an alternative to phone calls to communicate with them (*n* = 10). SMSs were considered the best method for recall appointments (*n* = 12), whereas a face-to-face communication would preferentially be used to remind the patient to be vaccinated against influenza (recommended vaccination) (*n* = 11).

## 4. Discussion

On the whole, this study shows a low level of patients’ interest in ICTs in LT management, mostly for the Internet and emails. The “no ICT” profile included most of the patients. Few patients and healthcare professionals had some expectations about ICTs, especially in specific conditions; phone call was the preferred ICT for reminders, transmission, and communication. Specific ICT-related profiles were identified (“no ICT”, “SMS”, “email” and “oral communication”), with the ultimate goal of identifying how we can optimize the use of ICTs to support effective communication and enhance LT therapeutic management by ICT intervention.

This is the first time that the preferences of patients and healthcare professionals about ICTs were studied in LT. This study is particularly relevant in the context of the burden of complex patients needing expensive and complex care, such as lung transplant patients. One of the strengths of this study relies on the comprehensive point of view on ICT preferences provided by the designed questionnaire. Questions are based, and already validated, on the literature review and clinical practices (protocols, communication, information flow). These questions were not designed as yes/no questions, and so the results must be viewed in light of this. However, we acknowledge that statistical power may have been hampered by the small number of well-characterized patients who were included in a single center and the small number of healthcare professionals from the framework of the French LT centers included in this analysis. Many other ICTs exist, but only these five ICT were chosen in the study because of their largest use [22,23,24,25]. Thus, most patients have knowledge and experience of these ICTs. Our LT center used mainly phone call for the transmission of information and communication, which might affect patients’ preferences and potentially bias the results. Their preferences might also differ from patients followed in other LT centers, because they have benefitted since transplantation from therapeutic education sessions in the framework of a program authorized by health authorities since 2008. However, all French LT centers have an LT coordinating nurse in particular to communicate with patients and have developed a common education program in the framework of the GETTAM. Furthermore, to complete the experience of patients with each of those technologies, it would have been interesting to use theories like Technology acceptance modeling (TAM) or the Unified Theory of Acceptance and Use of Technology (ATAUT) [35].

Surprisingly, our study highlights that 96% of patients and 90% of healthcare professionals would not like any reminder for daily drug intake, including immunosuppressive therapies. This result can probably be explained by the fact that patients benefit from therapeutic education, allowing them to identify the most appropriate method for their lifestyle to avoid forgetting medication and to respect the time of taking it (e.g., alarm on their mobile phone). Emails are a common method of communication, and their use in health services is increasing [22]. Mobile phones, including SMS, used as a support or as a reminder, have become an important tool for communication in health, and its applications are likely to increase over the years [37]; its usefulness has been proven in several chronic diseases. A significant reduction of graft rejection was shown with SMS reminders for taking immunosuppressive drugs delivered to pediatric recipients of liver transplants (or caregivers) [38]. An improvement in adherence by sending SMS reminders for taking daily treatments was highlighted in other diseases, such as asthma and human immunodeficiency virus infection [33,34]. Nevertheless, implementing ICTs in patients’ management may be more likely to be successful in some conditions. Indeed, 34% of our patients would appreciate reminders for taking punctual medications (such as pyrimethamine sulfadoxine or vitamin D), mainly by SMS or emails. This difference between daily and punctual drug intake is probably due to the difficulty of adherence in the latter case [9,10,11,12].

Adherence is also a challenge for all conditions other than medications. Some LT centers recommend that their lung transplant patients monitor daily pulmonary function by a spirometer at home to early identify potential complications. In our study, only 10% of patients and healthcare professionals would appreciate a reminder for breath measurement. This monitoring was considered either too restrictive or unnecessary for the interviewed patients. However, non-adherence was estimated from 2 to 62% [6,16]. SMS reminders incorporated in a spirometer were studied to improve adherence in breath home monitoring [39]. Our findings confirm that ICTs might be considered for all conditions to ensure a positive impact for patients’ prognosis.

The type of ICT may have to be considered in patients’ management according to the point of view of patients and healthcare professionals. Overall, face-to-face and phone call communication were preferred by both patients and healthcare professionals, especially to inform about any change in treatments and communicate together. The healthcare professionals appreciated emails for non-emergency situations and preferred to use SMS reminders for appointments and oral reminders for vaccination. A Cochrane review showed that SMS reminders, associated or not with sending postal mail, improve adherence at medical appointments compared to no reminder or postal mail alone, and they have a similar impact as a phone call with a lower cost for SMS [23]. This ICT was also chosen by patients for optimizing care; around 20% chose an SMS reminder for taking punctual medications, and almost 20% would appreciate an SMS as reminder for healthcare appointments. At the opposite end, the patients never chose the proposed answers of “website” and “smartphone application”. These ICTs showed a positive impact in the personalized adjustment of insulin doses among type 1 diabetic patients [32]. Our findings can be explained by a lack of knowledge of these methods or a lack of accessibility and visibility for their use in patients’ management.

Finally, our study highlighted, for the first time, that the use of ICTs may have to be adapted to the patients’ profile. The free statistical approach performed, the PCA, identified four ICT-related profiles. The specific value of ICT to specific groups of people is clear, despite the overall use of ICTs for health care communication being relatively low. Interestingly, the profile mainly composed of lung transplant patients for COPD, emphysema, or bronchiectasis and aged over 60-years-old did not choose ICTs, such as internet or email, compared to the profile composed of lung transplant patients for cystic fibrosis and aged under 40-years-old.

## 5. Conclusions

In conclusion, our study informs future development and implementation of ICTs to support effective communication in LT therapeutic management. ICT allows patients to manage their health information, communicate with healthcare providers, and participate actively in their healthcare. The type of ICT used may take into account the patient’s profile. Thus, our study provides practical guidance to enhance LT therapeutic management by ICT intervention. We have to keep in mind that more than a tool alone, the combination of strategies, including information, education by a multidisciplinary team, and reminders, is a promising approach to ensure the optimal management of our patients.

## Figures and Tables

**Figure 1 pharmacy-10-00075-f001:**
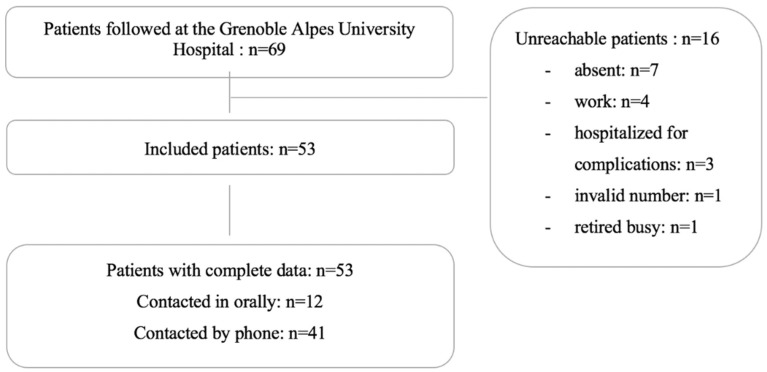
Flowchart of the lung transplant patient population from Grenoble Alpes University Hospital included in this analysis.

**Figure 2 pharmacy-10-00075-f002:**
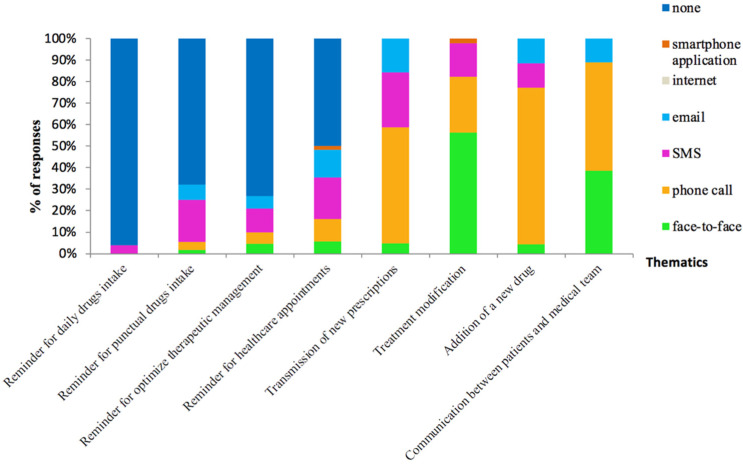
Patients’ preferences about ICT in their management and monitoring (*n* = 53).

**Figure 3 pharmacy-10-00075-f003:**
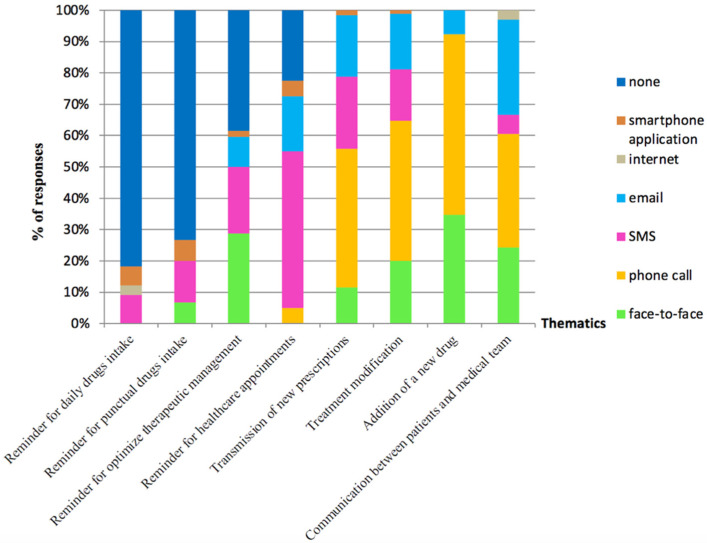
Perceptions of healthcare professionals about ICTs in therapeutic management and monitoring of lung transplant patients (*n* = 15).

**Table 1 pharmacy-10-00075-t001:** Socio-demographic and clinical characteristics of the lung transplant patient population included in this analysis (*n* = 53).

Characteristic		*n*	(%)
Sex	Male	33	−62
Age (years)	<40	15	−28
	40 ≤ Age ≤ 60	15	−28
	>60	23	−43
Marital Status	Married	26	−49
	Single	18	−34
	Divorced	9	−17
Occupational	Unemployed	19	−36
	Employed/Student	16	−30
	Retired	18	−34
ICT accessibility	Possession of a mobile phone	50	−94
	Package to send free SMS	37	−70
	Comfortable with SMS	41	−77
	Internet access at home	46	−87
	Comfortable with email	35	−66
	Comfortable with internet	39	−74
Indication of LT	COPD/Emphysema/bronchiectasis	28	−53
	Cystic fibrosis	14	−26
	Pulmonary fibrosis/PH	11	−21
Post-transplantation time	<1 year	12	−23
	≥1 year	41	−77
Immunosuppressive therapy regimen	tacrolimus + MMF + corticosteroids	26	−49
	tacrolimus + MMF + everolimus + corticosteroids	18	−34
	tacrolimus + AZA + corticosteroids	3	−6
	tacrolimus + everolimus + corticosteroids	3	−6
	tacrolimus + AZA + everolimus + corticosteroids	1	−2
	tacrolimus + MMF + everolimus	1	−2
	everolimus + corticosteroids	1	−2
Therapeutic Drug Monitoring	For 1 immunosuppressive drug	30	−57
	For 2 immunosuppressive drugs	23	−43

## Data Availability

The data presented in this study are available on request from the corresponding author. The data are not publicly available due to privacy.

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
