# Peer review of "Information and Communication Technologies in Lung Transplantation: Perception of Patients and Medical Teams"

_pharmacy, 2022, doi:10.3390/pharmacy10040075_

Round 1

Reviewer 1 Report

accept as is with minor english edits.

Author Response

We thank the Reviewer for his advice on grammar changes. We have taken them into account.

Reviewer 2 Report

Here the authors have shown the integration of different information and communication technologies in lung transplantation treatment to supervise and remind patients regarding their medicine intake and other aspects. They have valued the involvement of ICTs from the patients' and doctors' point-of-view. The study here conclusively proves that the ICT involvement will only help improve the treatment. Given that the world relies increasingly on technology, the seed for ICT research in medical monitoring systems is necessary.

I have a few quick questions for the authors.

1.    Patients' demography is wide enough; however, the sample number is meager for such a study. Please explain.

2.    The patient's thoughts on ICT involvement were recorded. Most patients do not prefer daily drug intake reminders (line 227). I am curious to know why which would help us change the situation.

3.    Are there any general FAQs for patients in smartphone applications and website login? If their concern is common, it can be cleared immediately rather than waiting for a medical professional's reply.

4.    To better bridge the gap between patients and ICTs, would it help create a rewarding situation? Say for proper medicinal intake for five months, 20% off on your next prescription, etc. Just another direction for the authors to think of.  

All the best!

Author Response

  1. Patients' demography is wide enough; however, the sample number is meager for such a Please explain.

We acknowledge that the sample number is small. We conducted a single-center study based on the total cohort of lung transplant patients followed at Grenoble Alpes Hospital during the study period. A multi-center study would have increased the sample size but would have been difficult to achieve due to the complexity of the interview. We discussed this point in the revised manuscript (page 7):“However, we acknowledge that statistical power may be hampered by the small number of well-characterized patients who were included in a single center, and the small number of healthcare professionals from the framework of the French LT centers included in this analysis.”

  1. The patient's thoughts on ICT involvement were recorded. Most patients do not prefer daily drug intake reminders (line 227). I am curious to know why which would help us change the situation.

The lung transplant patients followed at the University Hospital are patients who have to take many medications on a daily basis. They all receive therapeutic education including the importance of medication adherence. Patients are also made aware of the importance of the times they should take certain drugs such as immunosuppressants. Therapeutic education sessions enable them to find the method best suited to their lifestyle so that they do not forget to take their medication and to respect the times at which they should take it (e.g. alarm on the mobile phone). Daily medication reminders were therefore considered unnecessary by the patients. We discussed this point in the revised manuscript (page 8): “Surprisingly, our study highlights that 96% of patients and 90% of healthcare professionals wouldn’t like any reminder for daily drug intake, including immunosuppressive therapies. This result can probably be explained by the fact that patients benefit from therapeutic education allowing them to identify the most appropriate method for their lifestyle to avoid forgetting medication and to respect the time of taking it (e.g. alarm on the mobile phone).”

  1. Are there any general FAQs for patients in smartphone applications and website login? If their concern is common, it can be cleared immediately rather than waiting for a medical professional's reply.

We thank the reviewer for this relevant comment. There are FAQs from lung transplant patient associations or other French institutions. We have not specifically created a FAQ for lung transplant patients followed at the Grenoble Alpes University Hospital. So far, we have chosen to be available for patients in order to provide them with a personalized response to their needs/expectations. The idea suggested by the reviewer deserves to be considered as part of a new project to improve information for patients.

  1. To better bridge the gap between patients and ICTs, would it help create a rewarding situation? Say for proper medicinal intake for five months, 20% off on your next prescription, etc. Just another direction for the authors to think of.  

We thank the reviewer for this idea. Unfortunately, drugs prescribed for lung transplantation are fully reimbursed by the French health insurance. It is therefore impossible to propose such a reduction and contrary to French rules. The use of ICTs is intended to improve the management and therefore the prognosis of patients. The gap between patients and ICTs could be bridged by creating a fun game, in which the patient is his own character, to facilitate and generalize the use of ICTs.

Reviewer 3 Report

This well-written paper by Sébastien Chanoine, et al., entitled “Information and Communication Technologies in lung trans-2 plantation: perception of patients and medical team”. The authors highlight the use of information and communication technologies as a tool that would help in providing better therapeutic management that involves the diverse multidisciplinary team with regards to their lung transplant patient.  The demographic information in Table 1, related to the lung transplant patient population in this study is excellent with a remarkable response rate of 100%.  Patient preferred choices about information and communication technologies and their management and monitoring and perception of healthcare professionals, as shown in Figures 3 and 4, are easy to understand and are informative.

Author Response

This well-written paper by Sébastien Chanoine, et al., entitled “Information and Communication Technologies in lung trans-2 plantation: perception of patients and medical team”. The authors highlight the use of information and communication technologies as a tool that would help in providing better therapeutic management that involves the diverse multidisciplinary team with regards to their lung transplant patient.  The demographic information in Table 1, related to the lung transplant patient population in this study is excellent with a remarkable response rate of 100%.  Patient preferred choices about information and communication technologies and their management and monitoring and perception of healthcare professionals, as shown in Figures 3 and 4, are easy to understand and are informative.

We thank the Reviewer for his kinds comment on our manuscript.
